# The effect of lipid-lowering therapies on the pro-inflammatory and anti-inflammatory properties of vascular endothelial cells

Ewelina Woźniak●*, Marlena Broncel, Mateusz Niedzielski, Agnieszka Woźniak, Paulina Gorzelak-Pabiś

Department of Internal Diseases and Clinical Pharmacology, Laboratory of Tissue Immunopharmacology, Medical University of Lodz, Lodz, Poland

* ewelina.wozniak@umed.lodz.pl

**Data Availability Statement:** All relevant data are within the paper and its Supporting Information files.

## Abstract

Atherosclerosis and cardiovascular events can be prevented, or treated, using statin therapy, either alone or in combination with ezetimibe. Chronic inflammation, vascular proliferation, and the development of atherosclerosis are also influenced by 25-hydroxycholesterol (25-OHC). The aim of the study was to compare the direct pleiotropic effects of two commonly-used statins (atorvastatin, rosuvastatin), ezetimibe, and their combinations, on the mRNA expression of pro-inflammatory *IL1β*, *IL-18* and *IL-23* and anti-inflammatory *TGFβ*, *IL-35* (*EBI3*, *IL-12* subunits), *IL-10* and *IL-37*, in endothelial cells damaged by 25-OHC. It also analyzed IL-35 expression at the protein level. HUVECs were stimulated with atorvastatin (5 μM), rosuvastatin (10 μM), ezetimibe (1.22 μM), atorvastatin-ezetimibe (5 μM + 1.22 μM) or rosuvastatin-ezetimibe (10 μM + 1.22 μM), with or without pre-incubation with 10 μg/mL 25-OHC. mRNA expression was analyzed by real-time PCR. The protein level of IL-35 was analyzed by ELISA. In the pre-stimulated HUVECs, atorvastatin and rosuvastatin decreased mRNA expression of *IL1β*, *IL-18*, *IL-23*, *TGFβ*, *IL35* and increased mRNA expression of *IL-10* and *IL-37* compared to 25-OHC. Furthermore, only incubation with rosuvastatin and rosuvastatin-ezetimibe decreased IL-35 mRNA and protein levels. Ezetimibe down-regulated only *IL1β*. Treatment with rosuvastatin-ezetimibe and atorvastatin-ezetimibe reversed the effect of 25-OHC in *IL1β*, *IL-18* and *IL-35* mRNA expression. In conclusion, rosuvastatin has the strongest anti-inflammatory effects and is the best at reducing the effect of oxysterols. Both statins exert a greater anti-inflammatory effect than ezetimibe. The anti-inflammatory effect of the combination therapies appears to be based on the effects of the statins alone and not their combination with ezetimibe.

## 1. Introduction

The most common cause of death in high-income countries is cardiovascular disease. A key factor in its development is atherosclerosis, which is driven by chronic inflammation affecting all layers of the vascular wall, resulting in plaque formation and vascular obstruction. Vascular endothelial cells play a central role in inflammation, provoked by high levels of oxidized low-

**Funding:** The investigation was supported by statutory research granted for the Department of Internal Diseases and Clinical Pharmacology, Medical University of Lodz (number 503/5-165-01/503-51-001-19-00). The funders had no role in study design, data collection, and analysis, decision to publish, or preparation of the manuscript.

**Competing interests:** The authors have declared that no competing interests exist.

density lipoproteins (oxLDLs); this results in increased endothelial permeability, monocyte migration, and vascular smooth muscle cell (VSMC) migration into the intima, which contributes to the development of atherosclerosis [1].

One of the endogenous oxidative cholesterol derivatives involved in the atherosclerosis process is 25-hydroxycholesterol (25-OHC) [2]. It belongs to a subgroup of oxidized low-density lipoproteins that participate in chronic inflammation, vascular proliferation, and the development of atherosclerosis [3]. Elevated serum levels of 25-OHC are observed after high-fat meals and in patients with hypercholesterolemia [4–6]. 25-OHC exerts a pro-inflammatory effect on the vascular wall, resulting in endothelial dysfunction, increased oxidative stress, and pro-inflammatory cytokine production [7]. Furthermore, it has been shown that the inhibition of 25-OHC synthesis may inhibit atherosclerosis by suppressing macrophage foam cell formation [8]. Therefore, a proper understanding of the effect of 25-OHC on the vascular wall is crucial for atherosclerotic research.

The statins are cholesterol-lowering medications that inhibit the activity of 3-hydroxy-3-methyl-glutaryl-coenzyme A reductase. These drugs are commonly administered to treat atherosclerotic cardiovascular disease (CVD). The European Society of Cardiology (ESC) and the 2019 European Atherosclerosis Society (EAS) guidelines [9] recommend that in high and very high-risk patients, high-intensity atorvastatin (40–80 mg) or rosuvastatin (20–40 mg) therapy should be administered, and combined ezetimibe-statin therapy in cases of statin intolerance or insufficiency [10].

Effective antiatherosclerosis treatment plays a key role in avoiding cardiovascular incidents. For this reason, the present study compares the anti-inflammatory effects of clinically-used statins (atorvastatin and rosuvastatin), ezetimibe, and combined ezetimibe-statin in human vascular endothelial cells (HUVECs) damaged by oxidized cholesterol. A detailed analysis of the expression of pro-inflammatory *IL-1β*, *IL-18*, *IL-23*, and anti-inflammatory *IL35* (subunits *IL12A*, *EBI3*), transforming growth factor β (*TGF-β*), *IL-10* and *IL-37* was performed at the mRNA level. Additionally, IL-35 expression was determined at the protein level.

This is the first study to evaluate the immunomodulatory properties of atorvastatin, rosuvastatin and ezetimibe, and their combinations, in human umbilical vascular endothelial cells (HUVECs) after pre-stimulation with 25-OHC.

## 2. Results

### 2.1. Analysis of gene expression and protein levels

The experiment was performed over a period of 24 hours: four hours of pre-stimulation with 25-OHC + 20 hours of drug stimulation. Before the experiment, HUVECs were incubated in a medium for 24 hours. They were then pre-stimulated with 25-OHC (10 µg/mL) for four hours. Following this, the oxysterol was washed out and the cells were stimulated for an additional 20 hours with either atorvastatin (5 µM; 2793 ng/mL), rosuvastatin (10 µM; 4815 ng/mL), ezetimibe (1.22 µM; 500 ng/mL), or the following combinations: atorvastatin with ezetimibe or rosuvastatin with ezetimibe. Following 25-OHC pre-stimulation and drug stimulation, mRNA and the supernatants were isolated from HUVECs. The specific lengths of stimulation were selected based on previous observations of the integrity of the HUVECs in the RTCA-DP system [11]. Gene expression was measured vt reverse-transcription polymerase chain reaction in real time (RT-PCR) and protein levels were measured with enzyme-linked immunosorbent assay (ELISA).

## 2.2. Analysis of the expression of genes of pro-inflammatory cytokines: *IL-1β, IL-18, IL-23*

The administration of 25-hydroxycholesterol was associated with increased mRNA *IL1β* mRNA expression ($p < 0.001$). That expression was reduced by every subsequent stimulation with atorvastatin ($p < 0.01$), rosuvastatin ($p < 0.001$), ezetimibe ($p < 0.001$) or combined atorvastatin and ezetimibe ($p < 0.001$) or rosuvastatin and ezetimibe ($p < 0.05$) (Fig 1A).

The administration of 25-hydroxycholesterol was associated with an increase in *IL-18* mRNA ($p < 0.001$). However, these levels were reduced by subsequent addition of atorvastatin ($p < 0.001$), rosuvastatin ($p < 0.05$), atorvastatin-ezetimibe ($p < 0.01$) or rosuvastatin-ezetimibe ($p < 0.001$). Ezetimibe was not able to reverse the effect of 25-hydroxycholesterol ($p > 0.05$) (Fig 1B).

The administration of 25-hydroxycholesterol was associated with an increase in *IL-23* expression at the mRNA level ($p < 0.01$). This was reduced by atorvastatin ($p < 0.001$) or rosuvastatin treatment ($p < 0.001$). However, neither ezetimibe, atorvastatin-ezetimibe or rosuvastatin-ezetimibe were able to reverse the effect of 25-hydroxycholesterol ($p > 0.05$) (Fig 1C).

## 2.3. Analysis of the expression of genes of anti-inflammatory cytokines: (*TGFβ, IL-10, IL-37*)

25-hydroxycholesterol significantly decreased mRNA expression of *IL-10* and *IL-37* in HUVECs after 24 hours of culture as compared to unstimulated controls ($p < 0.05$). Atorvastatin ($p < 0.05$) and rosuvastatin ($p < 0.05$; $p < 0.001$), effectively increased these levels; however, no significant effect was observed for ezetimibe, ezetimibe-atorvastatin or ezetimibe-rosuvastatin ($p > 0.05$) (Fig 1D and 1E).

In addition, 25-hydroxycholesterol significantly increased mRNA expression of *TGFβ* in HUVECs after 24 hours of culture as compared to unstimulated control ($p < 0.001$). This expression was successfully decreased by every subsequent stimulation with atorvastatin ($p < 0.001$), rosuvastatin ($p < 0.001$), atorvastatin-ezetimibe ($p < 0.001$) or rosuvastatin-ezetimibe ($p < 0.001$); no change was observed for ezetimibe ($p < 0.05$) (Fig 1F).

## 2.4. Analysis of the expression of genes and proteins levels of IL-35 (IL-12, EBI3)

25-hydroxycholesterol increased *EBI3* mRNA level ($p < 0.001$). However, this was lowered by rosuvastatin ($p < 0.001$) and ezetimibe-rosuvastatin ($p < 0.01$). No significant effect was observed for ezetimibe, atorvastatin or ezetimibe-atorvastatin ($p > 0.05$) (Fig 2A).

25-hydroxycholesterol increased *IL-12A* mRNA level ($p < 0.001$). This effect was negated by atorvastatin ($p < 0.001$) and rosuvastatin ($p < 0.001$). In addition, ezetimibe-atorvastatin and ezetimibe-rosuvastatin were also able to decrease *IL-12A* gene expression, but the effect was weaker ($p < 0.05$). No significant effect was observed for ezetimibe ($p > 0.05$) (Fig 2B).

25-hydroxycholesterol increased IL-35 protein level ($p < 0.001$). This was negated by rosuvastatin ($p < 0.01$) and ezetimibe-rosuvastatin ($p < 0.001$). No significant effect was observed for atorvastatin, ezetimibe, or ezetimibe-atorvastatin ($p > 0.05$) (Fig 2C).

## 3. Discussion

This study is the first to investigate the effects of oxidized cholesterol (25-hydroxycholesterol) and statins (rosuvastatin, atorvastatin), and ezetimibe, alone and in combination, on pro-inflammatory and anti-inflammatory endothelial cell responses. The concentrations of atorvastatin (5 μM; 2793 ng/mL), rosuvastatin (10 μM; 4815 ng/mL) and ezetimibe (1.22 μM; 500 ng/mL) in the

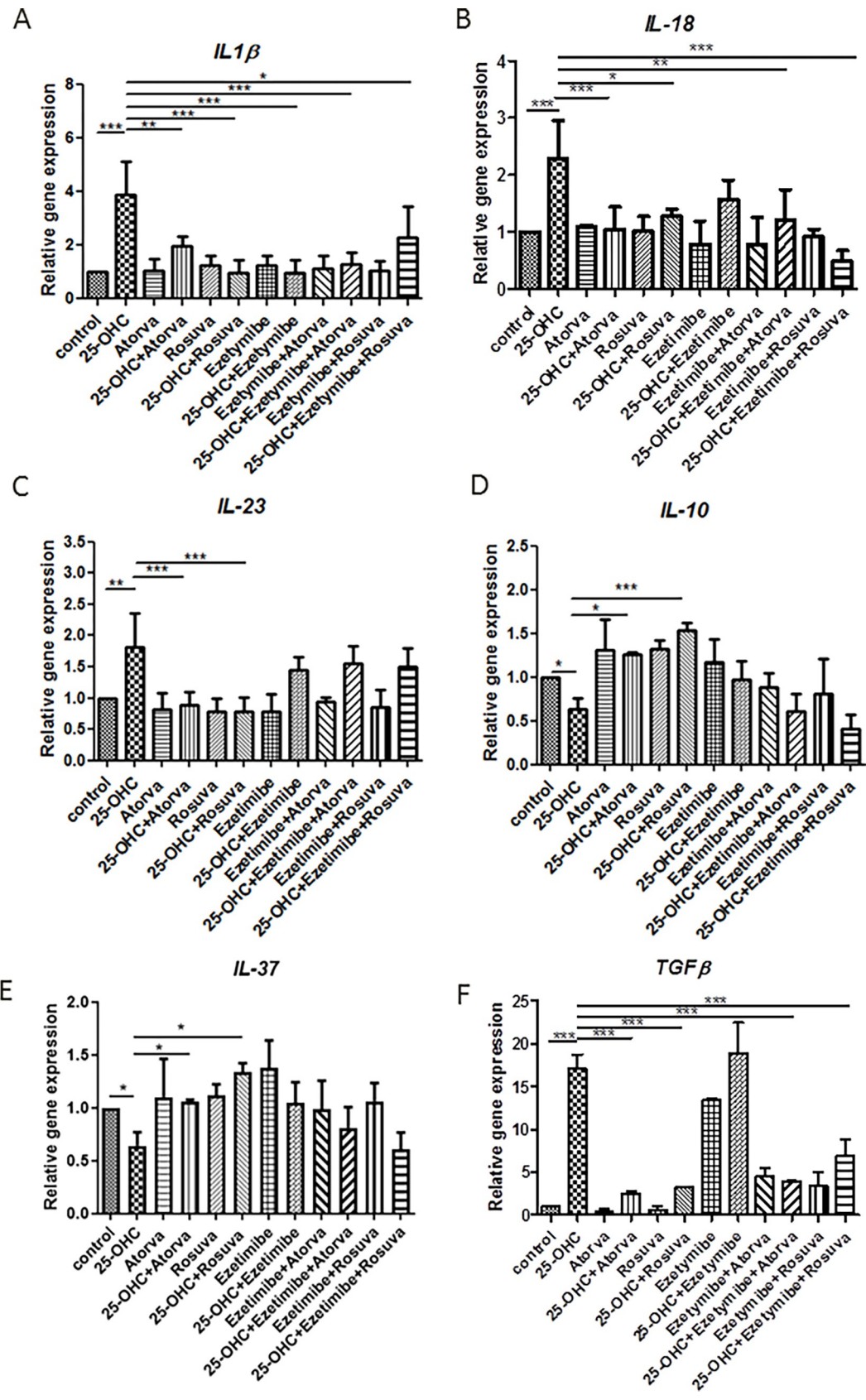

**Fig 1. Relative expression of genes (A-F) in HUVECs prestimulated with 25-OHC (10 μg/mL) for four hours, followed by removal of oxysterol and stimulation for another 20 hours (i.e.** A-F. 24 hours in total) with rosuvastatin (10 μM; 4815 ng/mL), atorvastatin (5 μM; 2793 ng/mL), ezetimibe (500 ng/mL), ezetimibe-rosuvastatin and ezetimibe-atorvastatin. Each experimental group was associated with a control group that was not pre-stimulated with 25-OHC. Mean ± SD calculated from four biological replicates. Significant differences from negative controls are indicated by $^*P < 0.05$; $^{**}P < 0.01$; $^{***}P < 0.001$. Statistical analysis was conducted using one-way ANOVA and the *post hoc* Tukey's test.

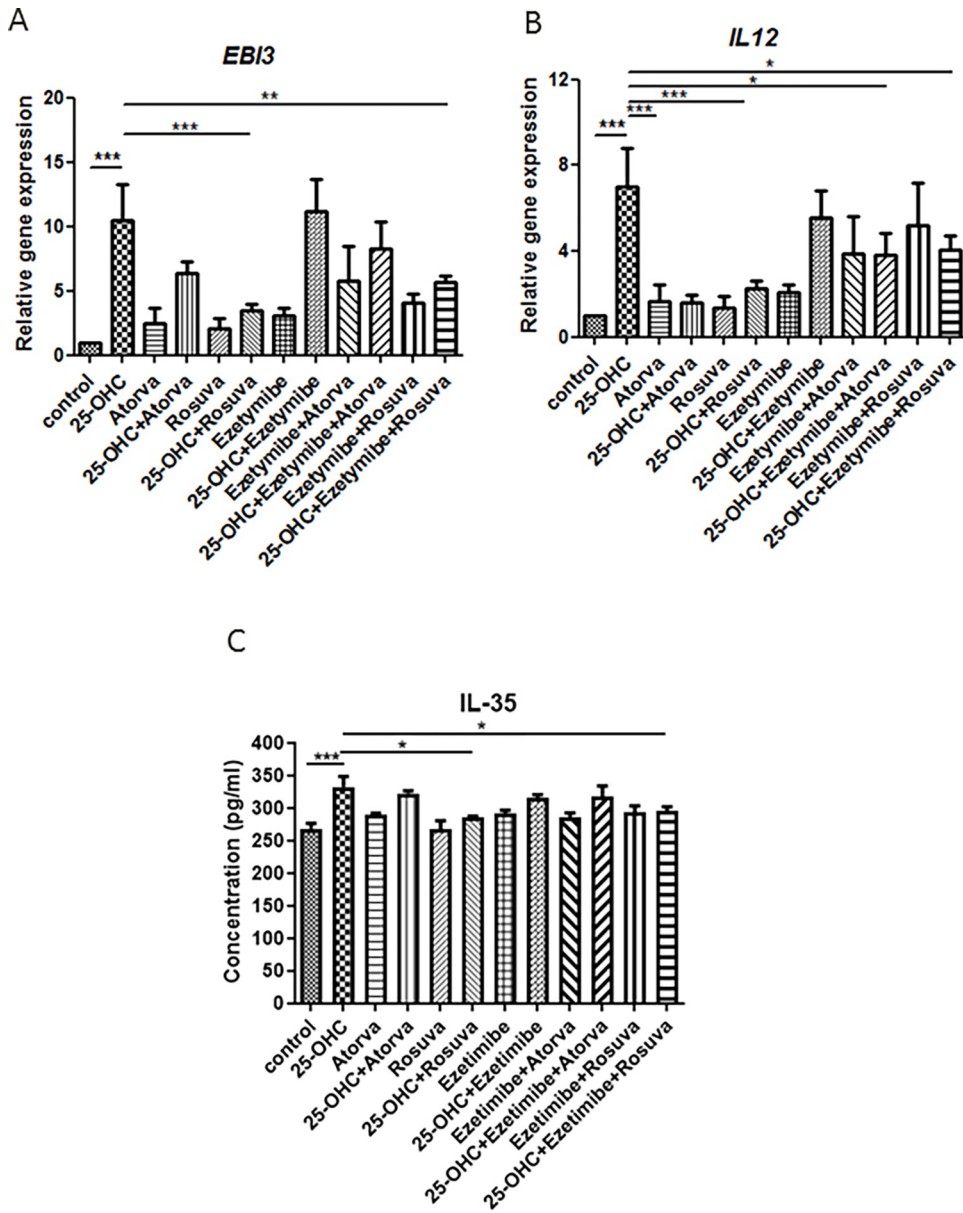

**Fig 2. Relative expression of gene *IL-35* (*EBI3*-A, *IL12*-B) and protein I in HUVECs prestimulated with 25-OHC (10 μg/mL) for four hours of incubation, followed by removal of oxysterol and stimulation for another 20 hours (i.e.** A-C. 24-hour incubation in total) with rosuvastatin (10 μM; 4815 ng/mL), atorvastatin (5 μM; 2793 ng/mL), ezetimibe (500 ng/mL), ezetimibe plus rosuvastatin and ezetimibe plus atorvastatin. Each experimental group has a control group without pre-stimulation of 25-OHC. Mean ± SD calculated from four biological replicates. Significant differences from negative controls are indicated by $^*P < 0.05$; $^{**}P < 0.01$; $^{***}P < 0.001$. Statistical analysis was conducted using one-way ANOVA and *post hoc* Tukey's test.

present study correspond to those in similar *in vitro* models [12]. They also correspond to those in the plasma of patients receiving the highest therapeutic doses of these drugs: 80 mg for atorvastatin [13], 40 mg for rosuvastatin [14], and 10 mg for ezetimibe [15, 16].

High-fat meals are rich in triglycerides (TG) and oxysterols such as 25-hydroxycholesterol, 7 a-hydroxycholesterol, 7 b-hydroxycholesterol, a-epoxycholesterol, b-epoxycholesterol and 7-ketocholesterol. These can impair endothelial function within three to four hours of ingestion, which is associated with peak postprandial lipemia [17]. In healthy people, plasma TG levels peak three to four hours after the ingestion of a high-fat meal, and tend to return to baseline within six to eight hours [18]. Therefore, in the present study, 25-OHC stimulation was stopped after four hours.

Inflammation is a very complex process that involves a range of endothelial factors and signaling pathways that regulate the chemotaxis, migration, and activation of leukocytes. The present study examines the key inflammatory factors that play significant roles in the endothelium [19]. Both IL-1β and IL-18 exert pro-inflammatory effects through the nuclear factor kappa-light-chain-enhancer of activated β cells (NF-κβ) pathway and are involved in the atherosclerosis process [20, 21]. It has previously been reported that IL-10 inhibits the proinflammatory activation of endothelial cells (EC) with respect to their stimulus and effector response. IL-10 was found to reduce the migration of monocytes and T lymphocytes across the endothelium and decrease endothelial production of chemokines following lipopolysaccharide (LPS) treatment [22]. IL-10 could suppress mitochondrial reactive oxygen species (mtROS), which partially mediate lysophosphatidylcholine (LPC)-induced EC activation and innate immune response [23]. IL-37 also prevents atherosclerotic plaque development [24]. Furthermore, extracellular IL-37 may influence the IL-18 pathway, as well as the adhesion and transmigration of neutrophils in human coronary artery endothelial cells (HCAECs) [25].

While IL-23 has strongly pro-inflammatory effects, IL-35 has been found to be a responsive anti-inflammatory cytokine; although it is upregulated during inflammatory responses, it is not a constitutively expressed housekeeping cytokine. IL-35 is induced by pro-inflammatory stimuli, and inhibits lipopolysaccharide (LPS)-induced endothelial cell activation [26, 27]. In turn TGF-β, which plays a key role in proper cell activity, is widely considered an anti-inflammatory cytokine [28]. It can attenuate inflammatory markers in VSMC and induce vascular remodeling by forming collagen and fibronectin in the extracellular matrix and promoting calcification. Atorvastatin and rosuvastatin have been found to have pleiotropic anti-inflammatory effects [10, 16]. Our findings indicate that both atorvastatin (5 μM) and rosuvastatin (10 μM) were able to negate the effect of 25-OHC (10 μg/mL) on HUVECs by decreasing *IL-1β*, *IL-18* and *IL-23* mRNA expression. These findings are consistent with literature data. Zhang *et al*. (2018) [29] report that atorvastatin treatment (5 mg/kg/day) was associated with a reduction of IL-1β in the serum of rabbits. Furthermore, Satoh *et al*. (2014) [30] note that rosuvastatin (0.4–10 μM) can also significantly reduce the levels of IL-1β and NLR family pyrin domain containing 3 (NLRP3) in human monocytic cell lines derived from an acute monocytic leukemia patient (THP-1 cells) subjected to oxidative stress. Chandrasekar *et al*. (2006) [31] report that atorvastatin (2.5–10 μM) can attenuate the deleterious effect of IL-18 on human coronary artery smooth muscle cells.

In addition, atorvastatin (5 mg/kg/day for 5 days) was found to significantly reduce IL-23 serum levels in mice with trinitrobenzene sulfonic acid (TNBS)- induced colitis [32]. Moreover, a randomized controlled trial on 159 patients with hypertension and carotid atherosclerosis showed that rosuvastatin (20 mg per day) can reduce IL-23 serum levels [33].

Our results indicate that atherogenic 25-OHC strongly increased the concentration of IL-35 in injured endothelial cells. In the case of IL-35, only atorvastatin was found to reduce *IL12A* mRNA levels following 25-OHC treatment. Rosuvastatin seems to offer more stable

effects by reducing the expression of two IL-35 subunits (*EBI3*, *IL12A*) and reducing the effect of oxysterol at the protein level. The observed reduction of *IL12A* mRNA by atorvastatin and rosuvastatin is in line with previous findings indicating that rosuvastatin (0.3 mg/kg/day; 21 days) can significantly reduce the serum level of IL-12 in mice with colitis induced by dextran sulfate sodium (DSS) [26, 34]. Furthermore, previous data suggests that IL-35 can suppress endothelial cell activation by inhibiting mtROS generation. Indeed, mtROS has also been found to contribute to inflammatory cytokine production and innate immune responses in macrophages and T cells [27].

Atorvastatin was found to inhibit dihydronicotinamide-adenine dinucleotide phosphate (NADPH)-oxidase dependent reactive oxygen species (ROS) generation; however, it showed no effect on mtROS generation, and activated IL-1β and IL-6 secretion in peripheral blood mononuclear cells (PBMC) from control and type 2 diabetes (T2D) patients. In turn, rosuvastatin mitigates coronary microembolization (CME)-induced cardiac injury by inhibiting NADPH-oxidase (Nox2)-induced ROS overproduction and alleviating apoptosis-related protein (p53/Bax/Bcl-2)-dependent cardiomyocyte apoptosis [35].

Recent studies indicate that IL-37 is able to inhibit the inflammatory response by suppressing the toll-like receptor 2 (TLR2)-NF-κB-intracellular adhesion molecule-1 (ICAM-1) pathway intracellularly in HCAECs; however, its effects and the mechanisms of inflammatory response in endothelial cells are not completely understood [25]. A key finding of our present study is that atorvastatin and rosuvastatin increased the expression of IL-37 in endothelial cells, which had been significantly decreased by 25-OHC. Limited data exists about the effect of statin or ezetimibe monotherapy on IL-37 expression. Our study is the first to describe the effect of such treatment, and of combined rosuvastatin-ezetimibe and atorvastatin-ezetimibe administration, on IL-37 mRNA expression in endothelial tissue. Shaoyuan *et al.* (2015) [36] report that atorvastatin administration (2 mg/kg/d for six weeks) resulted in higher IL-37 serum levels in rabbits with induced atherosclerosis compared to controls.

As previously mentioned, literature data suggests that TGFβ can play both anti- and pro-atherogenic roles [28]. Our findings indicate that both atorvastatin (5 μM) and rosuvastatin (10 μM) were able to negate the effect of 25-OHC (10 μg/mL) on HUVECs by decreasing TGFβ. Also, Kabel et al.(2017) report that rosuvastatin and/or ubiquinone treatment induced a significant decrease in serum creatine phosphokinase-MB (CK-MB), lactate dehydrogenase (LDH), troponin I, N-terminal-pro hormone BNP (NT-proBNP), tissue malondialdehyde (MDA), TGFβ and IL-6 in trastuzumab-treated mice [37]. The rosuvastatin-ubiquinone combination may represent a new therapeutic modality to ameliorate trastuzumab-induced cardiotoxicity: rats with an incomplete constriction of the abdominal aorta treated with rosuvastatin (2 and 4 mg/kg/day; five weeks) presented lower TGFβ protein levels than untreated rats [38]. Furthermore, atorvastatin treatment resulted in decreased vascular TGFβ levels and matrix metalloproteinase-2 (MMP-2) activity in renovascular hypertensive rats, thus ameliorating vascular remodeling [39].

Previous studies indicate that TGFβ plays a dual role in atherosclerosis, these being related to activation of the TGF-β–specific and bone morphogenetic protein (BMP)-specific signaling cascades in macrophages. The inhibition of MMP-2 expression by TGF-beta1 suggests that the latter, acting via SMAD family member 3 (Smad3), may promote plaque stability [40]. Nurgazieva et al. [41] suggest that the effect of TGF-β on atherosclerosis depends on the balance between Smad1/5- and Smad2/3-dependent signaling, where Smad1/5 is pro-atherogenic, and Smad2/3 has anti-atherogenic effects. However, endothelial TGFβ signaling is one of the primary drivers of atherosclerosis-associated vascular inflammation. Chen et al. [42] indicate that inhibition of endothelial TGFβ signaling in hyperlipidemic mice reduces vessel wall inflammation and vascular permeability, resulting in the arrest of disease progression and regression of

established lesions. These pro-inflammatory effects resulting from endothelial TGFβ signaling stand in stark contrast with its effects in other cell types, and show the potential of cell-type specific therapeutic intervention aimed at controlling atherosclerosis.

Unlike statins, the broad pleiotropic effect of ezetimibe on a number of anti-inflammatory mechanisms has not been confirmed. Undoubtedly, while ezetimibe is effective at reducing LDL cholesterol, it is more beneficial when used as an adjunct to statins than as monotherapy [43]. Krysiak et al. suggest that insulin-resistant patients with hypercholesterolemia and high cardiovascular risk may benefit the most from combined simvastatin and ezetimibe treatment [44]. It has also been found that combined use of statin and ezetimibe can further reduce low-density lipoprotein cholesterol (LDL-C), total cholesterol and triglyceride levels when compared with monotherapy [29]. Also, some studies suggest it exerts a protective effect on endothelial cells against oxLDL [16]. Our present findings indicate that ezetimibe (500 ng/mL) can only decrease the IL-1β mRNA level following 25-OHC (10 μg/mL) stimulation; no similar effect was observed for the other analyzed genes. Our findings are in line with literature data.

Suchy et al. (2014) [45] report that ezetimibe (22 ng/mL) treatment reduced IL-1β expression in macrophages pre-stimulated with LPS (1 μg/mL). The cells were taken from patients with hypercholesterolemia.

Similarly to our present findings, Tie et al. (2015) [46] report that ezetimibe has no effect on TGFβ: Apolipoprotein E (ApoE)$^{-/-}$ mice fed a diet rich in saturated fat and treated with ezetimibe (10 mg/kg/day; 28 days) showed no changes in serum concentration of TGF-β compared to untreated mice.

Our study shows that ezetimibe-atorvastatin or ezetimibe-rosuvastatin were able to significantly negate the deleterious effect of 25-OHC on the mRNA expression of *IL-1β*, *IL18*, *IL35 (IL12A)*, and *TGFβ*. In addition, these combination therapies did not appear to have any superiority over single statin monotherapy for inflammatory processes. The anti-inflammatory effect of the combination therapies appears to be due to the effects of the statins alone and not to their combination with ezetimibe. This is the first study to compare those therapies on human endothelial cells.

Previous studies comparing the effects of statin and ezetimibe therapy on inflammatory processes have mainly been restricted to clinical trials on patients. In a study of the systemic anti-inflammatory and endothelial protective effects of ezetimibe (10 mg/day; 90 days), atorvastatin (40 mg/day; 90 days) and atorvastatin-ezetimibe (in the same doses) in patients with hypercholesterolemia, Krysiak et al. (2012) report that atorvastatin and combined therapy had beneficial effects on high-sensitivity C-reactive protein (hsCRP), ICAM-1, tumor necrosis factor α (TNFα), interferon γ (IFNγ), and IL-2 serum concentrations, while ezetimibe monotherapy had no effect [47, 48].

For comparison, Oh et al. (2020) [35] examined the effect of rosuvastatin (20 mg/day) and rosuvastatin with ezetimibe (5 mg + 10 mg/day) on carotid plaque inflammation, as assessed by positron emission tomography ($^{18}$FDG PET/CT) in patients with acute coronary syndrome. Both treatments significantly reduced inflammation, but with no differences between the groups. Also, the lipid levels did not change over the course of therapy. It should be emphasized that while this study compared high-dose rosuvastatin and low-dose rosuvastatin-ezetimibe, it did not test the same doses of rosuvastatin.

The main limitation of our work is that it is not possible to directly translate our *in vitro* findings into the process of atherosclerosis taking place *in vivo*. Our research suggests that, under these specific experimental conditions, combined ezetimibe-statin therapy does not further decrease the gene expression of pro-inflammatory cytokines in endothelial cells following 25-OHC stimulation *in vitro*; however, this does not mean that these combinations do not offer benefits to other tissues or cells associated with arteriosclerosis formation *in vivo*.

Ezetimibe inhibits the absorption of cholesterol from the small intestine, leading to a reduction in intestinal cholesterol transmission to the liver, which protects the vascular endothelium from the inflammatory and damaging effects of cholesterol *in vivo* [43]. Although our study showed that ezetimibe treatment yields minimal anti-inflammatory effects and does not appear to enhance the anti-inflammatory effects of statins when added in combination, its clinical benefits remain unclear. Therefore, further studies are needed to compare statin monotherapy with combined statin-ezetimibe therapy *in vivo*.

## 4. Conclusion

Treatment with 25-hydroxycholesterol can induce inflammation in endothelial cells. Of the tested statins, rosuvastatin has the strongest anti-inflammatory effect and most effectively reduces the effect of oxysterols. Both atorvastatin and rosuvastatin exert a greater anti-inflammatory effect than ezetimibe. Although the tested combination therapies bestow anti-inflammatory effects, these could be due to the statins alone and not to the combination with ezetimibe. Therefore, they appear to have no advantage over statin monotherapy.

## 5. Materials and methods

### 5.1. Chemicals

HUVECs, trypsin with ethylenediaminetetraacetic acid (EDTA), trypsin neutralizing solution, endothelial cell growth medium-2 (EGM-2) with hydrocortisone, human fibroblastic growth factor-β (hFGF-β), vascular endothelial growth factor (VEGF), human recombinant E.coli, lyophilized powder (R3-IGF-1), ascorbic acid, human epidermal growth factor (hEGF), gentamicin (GA-1000), heparin and fetal bovine serum (FBS), were purchased from Lonza (Switzerland). Atorvastatin, rosuvastatin, ezetimibe, 25-hydroxycholesterol (25-OHC) and primers were bought from Sigma-Aldrich (USA). RNeasy Mini Kit was bought from Qiagen (Germany). High-Capacity complementary DNA (cDNA) Kit and SYBR-Green PCR Mastermix were purchased from Applied Biosystems (USA). Other chemicals were purchased from Roth (Germany) and POCh (Poland) and were of analytical grade.

### 5.2. Cell culture in monolayers

HUVECs were cultured in EGM-2 containing 10% FBS, hydrocortisone, hFGF-B, VEGF, R3-IGF-1, ascorbic acid, hEGF, GA-1000, heparin and penicillin (100 U/ml), and streptomycin (100 μg/ml) at 37˚C, 5% $CO_2$.

Following trypsinization, the HUVECs were separately seeded on 24-well plates at a density of 100,000 cells per well in a 600 μl EGM-2. After reaching 80–90% confluence, the HUVECs were stimulated with 25-hydroxycholesterol (10 μg/ml) for four hours. After incubation, the cells were centrifuged, the compound was discarded, and the HUVECs were stimulated with atorvastatin (5 μM), rosuvastatin (10 μM) and ezetimibe (500 ng/ml) for 20 hours. Total cell stimulation was 24 hours: four hours of pre-incubation with 25-OHC + 20 hours of drug incubation. After incubation, the cells were centrifuged, the compounds were discarded, and the cells were resuspended in EGM-2 medium.

No significant changes in HUVEC cell viability were observed as a result of treatment with statins and ezetimibe at the selected concentrations, as previously presented by Niedzielski et al. [11]. In each series of probes, cell viability was determined by the Trypan Blue dye exclusion test.

The investigations were approved by the Bioethics Committee of the Medical University of Lodz No. RNN/363/19/KE).

### 5.3. Gene expression

mRNA were isolated after 20 hours of drug incubation, which corresponds to 24 hours of the experiment (four hours of pre-incubation with 25-OHC + 20 hours of drug incubation). RNA was extracted with the RNeasy Mini Kit (Qiagen, Germany), following the manufacturer's instructions. Potential genomic DNA contamination was removed by on-column DNase I digestion. cDNA synthesis was performed using High-Capacity cDNA Kit (Applied Biosystems, USA). The cDNA was quantified by real-time PCR using SYBR-Green PCR Mastermix and was purchased from Applied Biosystems (USA). *EF1α* (elongation factor *α)* was amplified as a housekeeping gene. The reaction was conducted as follows: four minutes at 95˚C, followed by 40 cycles of 15 s at 95˚C and 60 s at 60˚C. The $2^{\Delta Ct\ (Ctgene-CtEF1\text{-}\alpha)}$ method was used to calculate the expression levels of the studied genes [49]. The primers were designed utilizing Primer-BLAST NCBI—NIH website: https://www.ncbi.nlm.nih.gov/tools/primer-blast/ (S1 Table).

### 5.4. Enzyme-linked immunosorbent assay (ELISA) for IL-35

The supernatants were isolated after four hours of pre-incubation with 25-OHC + 20 hours of drug incubation. The IL-35 levels in the HUVEC supernatants was assessed by enzyme-linked immunosorbent assay (ELISA) using an ELISA ST-360 microplate reader (450nm and 630nm) according to the manufacturer's protocol (Cloud-Clone, Katy, USA). The range of IL-35 was 1.56–300 pg/mL.

### 5.5. Statistical analysis

The distribution of particular variables was verified by the Shapiro-Wilk W-test, and homogeneity of variance was determined with the Brown–Fisher test. The results are presented as mean ± SD for variables with normal distribution. Statistical analysis was conducted using the one-way analysis of variance (ANOVA) followed by Tukey's *post hoc* multiple comparisons procedure. The difference was considered to be significant for $P<0.05$. Each analysis was performed in four independent experiments, with each experiment being repeated three times. All analyses were conducted with the STATISTICA 13.1 software (2000 Stat-Soft, Inc., Tulsa, USA).

## Supporting information

**S1 Table. The primers used in the real-time PCR.**
(DOCX)

## Author Contributions

**Conceptualization:** Ewelina Woźniak, Marlena Broncel, Paulina Gorzelak-Pabiś.

**Formal analysis:** Ewelina Woźniak.

**Funding acquisition:** Marlena Broncel.

**Investigation:** Mateusz Niedzielski, Agnieszka Woźniak.

**Methodology:** Ewelina Woźniak, Mateusz Niedzielski, Agnieszka Woźniak, Paulina Gorzelak-Pabiś.

**Project administration:** Ewelina Woźniak.

**Supervision:** Ewelina Woźniak, Marlena Broncel, Paulina Gorzelak-Pabiś.

**Visualization:** Ewelina Woźniak.

**Writing – original draft:** Ewelina Woźniak.

**Writing – review & editing:** Ewelina Woźniak, Marlena Broncel, Paulina Gorzelak-Pabiś.

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
