## [Decision Letter · Decision Letter 0]

15 Jul 2022

PONE-D-22-07586Effect of lipid-lowering therapies on pro-inflammatory and anti-inflammatory properties of vascular endothelial cellsPLOS ONE

Dear Dr. Woźniak 

Thank you for submitting your manuscript to PLOS ONE. After careful consideration, we feel that it has merit but does not fully meet PLOS ONE’s publication criteria as it currently stands. Therefore, we invite you to submit a revised version of the manuscript that addresses the points raised during the review process.

We look forward to receiving your revised manuscript.

Kind regards,

Xianwu Cheng, M.D., Ph.D., FAHA

Academic Editor

PLOS ONE

Journal Requirements:

The investigation was supported by statutory research granted for the Department of Internal Diseases and Clinical Pharmacology, Medical University of Lodz (number 503/5-165-01/503-51-001-19-00). The funders had no role in study design, data collection, and analysis, decision to publish, or preparation of the manuscript. The specific roles of these authors are articulated in the ‘author contributions’ section.

5. Thank you for submitting the above manuscript to PLOS ONE. During our internal evaluation of the manuscript, we found significant text overlap between your submission and the following previously published works, some of which you are an author.

- https://journals.plos.org/plosone/article?id=10.1371%2Fjournal.pone.0256996

Please revise the manuscript to rephrase the duplicated text, cite your sources, and provide details as to how the current manuscript advances on previous work. Please note that further consideration is dependent on the submission of a manuscript that addresses these concerns about the overlap in text with published work.

Reviewers' comments:

Reviewer's Responses to Questions

**Comments to the Author**

1. Is the manuscript technically sound, and do the data support the conclusions?

Reviewer #1: Yes

Reviewer #2: Partly

2. Has the statistical analysis been performed appropriately and rigorously? 

Reviewer #1: Yes

Reviewer #2: Yes

3. Have the authors made all data underlying the findings in their manuscript fully available?

Reviewer #1: Yes

Reviewer #2: Yes

4. Is the manuscript presented in an intelligible fashion and written in standard English?

Reviewer #1: No

Reviewer #2: No

5. Review Comments to the Author

Reviewer #1: The present study examined the expression profiles of the effects of statins, ezetimibe, and their combinations on 25-hydroxycholesterol (25-OHC)-induced inflammatory and anti-inflammatory cytokines in endothelial cells (ECs). The authors suggested that there is no advantage of combination therapy with rosuvastatin and ezetimibe over rosuvastatin alone in the inflammatory process in atherosclerosis. However, the present study lacks novelty as most of the findings had been reported by other researchers (PMID: 33725908). For example, one study showed that ezetimibe therapy improved the systemic anti-inflammatory effects of simvastatin in patients with isolated hypercholesterolemia (PMID: 24354929). Another study demonstrated that combined use of statin and ezetimibe can further reduce the low-density lipoprotein cholesterol (LDL-C), total cholesterol and triglyceride levels when compared with the monotherapy (PMID: 30480766). Therefore, this reviewer has several following concerns and suggestions:

Major

1. IL-35 is widely regarded as an anti-atherogenic and anti-inflammatory cytokine in atherosclerosis (PMID: 31731100; PMID: 28648331; PMID: 29371247). In the abstract and discussion (line 18, line 164), the authors stated that IL-35 was one of the anti-inflammatory cytokines that were studied. However, in the introduction (line 64), the authors indicated that IL-35 was the pro-inflammatory cytokine, which was inconsistent with others’ results. Besides, this study showed that 25-OHC increased the levels of mRNA (IL12A and EBI3) and protein of IL-35, and this effect was inhibited by rosuvastatin. What is the possible interpretation of these results? The authors need to expand the discussion.

2. Since TGF-β signaling is a crucial contributor in endothelial to mesenchymal transition (EndMT), a process involved in the progression of atherosclerosis. One study (PMID: 34157851) indicates that statins may have inhibitory effects on EndMT in vivo. In this study, 25-OHC increased the expression of TGF-β in HUVECs, while atorvastatin, rosuvastatin and the combination of atorvastatin and ezetimibe decreased this effect. This is contradictory to the anti-inflammatory effect of statins. Whether the effect of atorvastatin and rosuvastatin on TGF-β expression is independent of the anti-inflammatory properties? The authors should discuss it further.

3. Previous studies showed that the dual roles of TGFβ in atherosclerosis have been attributed to activation of TGF-β–specific and BMP-specific signaling cascades in macrophages (PMID: 10825169; PMID: 25505291). However, endothelial TGFβ signaling promotes vascular inflammation and atherosclerosis (PMID: 31572976). These contradictory studies indicate that TGFβ is not suitable to be an anti-inflammatory marker in ECs. I suggest that the authors use IL-10 as an anti-inflammatory cytokine in ECs. IL-10 possesses multiple anti-atherogenic activities, including blocking atherogenic lysophosphatidylcholine-induced EC activation (PMID: 31731100) and reducing intercellular adhesion molecule 1 (ICAM-1) expression in ECs (PMID: 12223518).

4. In the method, HUVECs were treated with oxysterol for 4 hours, then washed out and the cells were stimulated with the corresponding drugs. Is there any reason to wash oxysterol out before addition of a drug under study? The authors shall co-incubate cells with oxysterol and other drugs throughout the duration of co-treatment.

Minor

1. It should be highlighted that this manuscript requires rigorous editing by someone with competence in technical English grammar and sentence structure so that the aims and results are clear to the reader.

Reviewer #2: PLOS One Review 07/12/22 - PONE-D-22-07586

This article presents cytokine gene expression data and IL-35 concentrations from HUVECs stimulated with 25-OH-cholesterol that were treated or not with statins and/or ezetimibe. The authors conclude that there is no advantage of adding ezetimibe to statins over statin monotherapy. The manuscript addresses an important topic of interest. However, based on the in vitro experiments presented in this manuscript, this conclusion does not seem to be justified. The fact that ezetimibe in combination with statins under the specific experimental conditions does not further decrease the gene expression of pro-inflammatory cytokines in HUVECs, does not proof that such combinations are without any benefits to other tissues and cells relevant for arteriosclerosis formation. Ezetimibe inhibits intestinal cholesterol absorption, which in vivo prevents the inflammatory and damaging effects of cholesterol on vascular endothelium. While this study demonstrates that the anti-inflammatory effects of ezetimibe are minimal and do not enhance the anti-inflammatory effects of statins when added in combination, no statements regarding clinical benefits can be drawn from this study. The conclusions therefore should be changed accordingly, and the limitations of this in vitro study should be emphasized in the Discussion section of this manuscript. Furthermore, the study from Oh et al. (reference 35) compares high-dose rosuvastatin vs low-dose rosuvastatin combined with ezetimibe. Therefore, no group differences might indicate an advantage of the combination therapy, since lower doses may reduce adverse treatment effects. The statement in lines 248 to 250 seems therefore incorrect.

The Method description is unclear and inconsistent with the statements in the Abstract and Results section of the manuscript. For example, the duration of culture of HUVECs varies throughout the manuscript between 12 and 24 hours. Furthermore, the number of repetitions of the experiments remains unclear. And the current manuscript version contains a number of grammar and spelling errors. Abbreviations are not consistently defined upon their first use in the manuscript.

IL-35 is called anti-inflammatory in the abstract, but pro-inflammatory in the introduction and other parts of the manuscript.

Lines 77-79: “mRNA and supernatants were isolated after the 20 hours of drug stimulation. mRNA and supernatants were then isolated from HUVECs after 25-hydroxycholesterol (25-OHC) pre-stimulation and drug stimulation.” This seems to be duplicate statements.

Line 94, similar lines 97-98: “…was added after 25-OHC prestimulation reduced the IL-18 mRNA level”. This seems to be a grammar error.

Line 138: “protine”. Please change to “protein”.

Please add a reference for the statement in lines 154-156

Lines 162-164: “Further, the role of IL-37 may be related to the IL-18 pathway extracellularly and involved in the adhesion and transmigration of neutrophils in human coronary artery endothelial cells (HCAECs).” Please correct the sentence structure.

Lines 220: “for other analyzing genes”. Did you mean “for the other analyzed genes”?

Lines 239-240: ‘aniinflammatory effect”. Please correct to “anti-inflammatory effect”

Line 241: What does the abbreviation “hsCRP” stand for?

Line 252: “Our study shows that 25-hydroxycholesterol can damage the endothelium …”. Cytokine gene expression changes alone do not proof any endothelial damage.

Line 263: “purchased in Lonza”. Please change to “purchased from Lonza”. Similar errors were repeated several times in the manuscript text.

Line 265: “Rneasy Mini Kit”. Please change to “RNeasy Mini Kit”.

Line 270: “Cells treatment in the viability level and gene expression” This rephrase the title.

Line 279: What is “proper medium”?

Could the authors pleased explain why 25-hydroxycholesterol was removed prior to treatment with statins and/or ezetimibe, since in vivo high cholesterol levels would be present throughout the treatment with these drugs.

Line 292: “G Gene expression”. Did the authors mean “Gene expression”?

Line 301: Livak and Schmittgen 2001: This reference is not included in the reference list.

Line 317: “Shapiro-Will test”. Please correct to “Shapiro-Wilk test”.

Lines 320-322: “The individual analysis was performed in nine-four independent experiments, while each experiment was repeated twice or three times depending on the method.” This description seems unclear.

The information in the Acknowledgment section may belong to the Funding information section.

6. PLOS authors have the option to publish the peer review history of their article (what does this mean?). If published, this will include your full peer review and any attached files.

Reviewer #1: No

Reviewer #2: No

---

## [Author Response · Author response to Decision Letter 0]

26 Sep 2022

Dear Academic Editor 

Professor Xianwu Cheng, 

We are very grateful to the all Reviewers for thorough reading of our paper and the indicated comments, which significantly helped to improve the quality of the presented manuscript. We appreciate very much the positive feedback from the reviewers and the offered chance to send the revised version of our manuscript to PLOS ONE. We have corrected the paper in the line with Reviewers remarks and we hope that it has become more clear now. English of the manuscript was carefully checked and corrected by an English native speaker (please see the attached certificate). 

We have made an improvement:

- ‘Funding Information’ and ‘Financial Disclosure’

- Acknowledgments section

- I included amended statements: The investigation was supported by statutory research granted for the Department of Internal Diseases and Clinical Pharmacology, Medical University of Lodz (number 503/5-165-01/503-51-001-19-00). The funders had no role in study design, data collection, and analysis, decision to publish, or preparation of the manuscript.

- After accepting the manuscript, I will send you the raw data file

- I revised the manuscript to rephrase the duplicated text

 Sincerely yours, 

Ewelina Woźniak, Corresponding Author

 

The suggestions and questions raised by Reviewer #1

The present study examined the expression profiles of the effects of statins, ezetimibe, and their combinations on 25-hydroxycholesterol (25-OHC)-induced inflammatory and anti-inflammatory cytokines in endothelial cells (ECs). The authors suggested that there is no advantage of combination therapy with rosuvastatin and ezetimibe over rosuvastatin alone in the inflammatory process in atherosclerosis. However, the present study lacks novelty as most of the findings had been reported by other researchers (PMID: 33725908). For example, one study showed that ezetimibe therapy improved the systemic anti-inflammatory effects of simvastatin in patients with isolated hypercholesterolemia (PMID: 24354929). Another study demonstrated that combined use of statin and ezetimibe can further reduce the low-density lipoprotein cholesterol (LDL-C), total cholesterol and triglyceride levels when compared with the monotherapy (PMID: 30480766).

We thank the Reviewer for this valuable suggestion. We changed the conclusions and the limitations of this study were emphasized in the Discussion section of this manuscript. The statement in lines 248 to 250 has been removed.

We added the below information in the manuscript: 

Discussion

“Unlike statins, the broad pleiotropic effect of ezetimibe on a number of anti-inflammatory mechanisms has not been confirmed. Undoubtedly, ezetimibe is effective at reducing LDL cholesterol. It is more beneficial when used as an adjunct to statins than as monotherapy [43]. Krysiak et al. suggest that insulin-resistant patients with hypercholesterolemia and high cardiovascular risk may benefit the most from combined simvastatin and ezetimibe treatment [44]. It has also been found that combined use of statin and ezetimibe can further reduce low-density lipoprotein cholesterol (LDL-C), total cholesterol and triglyceride levels when compared with monotherapy [29].” (lines 261-268)

”The main limitation of our work is that it is not possible to directly translate our in vitro findings into the process of atherosclerosis taking place in vivo. Our research suggests that, under these specific experimental conditions, ezetimibe in combination with statins does not further decrease the gene expression of pro-inflammatory cytokines in endothelial cells stimulated by 25-OHC in vitro; however, this does not mean that such combinations are without any benefits to other tissues and cells relevant for arteriosclerosis formation in vivo. Ezetimibe inhibits the absorption of cholesterol from the small intestine, leading to a reduction in intestinal cholesterol transmission to the liver, which protects the on vascular endothelium from the inflammatory and damaging effects of cholesterol in vivo [43]. Although our study showed that the anti-inflammatory effects of ezetimibe were minimal and did not enhance the anti-inflammatory effects of statins when added in combination, it is not possible to draw firm statements regarding clinical benefits. Therefore, further studies are needed to compare statin monotherapy with statin-ezetimibe combined therapy in vivo.” (lines 302-314).

Major

1. IL-35 is widely regarded as an anti-atherogenic and anti-inflammatory cytokine in atherosclerosis (PMID: 31731100; PMID: 28648331; PMID: 29371247). In the abstract and discussion (line 18, line 164), the authors stated that IL-35 was one of the anti-inflammatory cytokines that were studied. However, in the introduction (line 64), the authors indicated that IL-35 was the pro-inflammatory cytokine, which was inconsistent with others’ results. Besides, this study showed that 25-OHC increased the levels of mRNA (IL12A and EBI3) and protein of IL-35, and this effect was inhibited by rosuvastatin. What is the possible interpretation of these results? The authors need to expand the discussion.

We thank the Reviewer for this valuable suggestion. This introductory paragraph was rephrased.

“A detailed analysis of the expression of pro-inflammatory IL-1�, IL-18, IL-23, and anti-inflammatory IL35 (subunits IL12A, EBI3), transforming growth factor β (TGF-�), IL-10 and IL-37 was performed at the mRNA level. Additionally, IL-35 expression was determined at the protein level. “ (lines 62-66).

We added in the manuscript this information: 

Discussion

”Previous studies indicate that IL-35 is a responsive anti-inflammatory cytokine, which is upregulated during inflammatory response, but not a constitutively expressed housekeeping cytokine. IL-35 is not constitutively expressed in tissues, being induced by pro-inflammatory stimuli, and inhibits lipopolysaccharide (LPS)-induced endothelial cell activation [27]. ” (lines 179-183).

”Furthermore, previous data suggest that IL-35 can suppress endothelial cell activation by inhibiting mtROS generation. Indeed, mtROS is known to contribute to inflammatory cytokine production and innate immune responses in macrophages and T cells [27]. (lines 211-214).

”Atorvastatin was found to inhibit dihydronicotinamide-adenine dinucleotide phosphate (NADPH)-oxidase dependent reactive oxygen species (ROS) generation; however, it showed no effect on mtROS generation, and activated IL-1β and IL-6 secretion in peripheral blood mononuclear cells (PBMC) from control and type 2 diabetes (T2D) patients. In turn, rosuvastatin mitigates coronary microembolization (CME)-induced cardiac injury by inhibiting NADPH-oxidase (Nox2)-induced ROS overproduction and alleviating apoptosis-related protein (p53/Bax/Bcl-2)-dependent cardiomyocyte apoptosis [35]. ’’ (lines 215-221).

Complete the literature reference:

[27] Li X, Shao Y, Sha X, Fang P, Kuo YM, Andrews AJ, et al. IL-35 (interleukin-35) suppresses endothelial cell activation by inhibiting mitochondrial reactive oxygen species-mediated site-specific acetylation of H3K14 (histone 3 lysine 14). Arterioscler Thromb Vasc Biol. 2018;38: 599–609. doi:10.1161/ATVBAHA.117.310626

[35] Cao Y, Chen Z, Jia J, Chen A, Gao Y, Qian J, Ge J. Rosuvastatin Alleviates Coronary Microembolization-Induced Cardiac Injury by Suppressing Nox2-Induced ROS Overproduction and Myocardial Apoptosis. Cardiovasc Toxicol. 2022 Apr;22(4):341-351. doi: 10.1007/s12012-021-09716-4.

2. Since TGF-β signaling is a crucial contributor in endothelial to mesenchymal transition (EndMT), a process involved in the progression of atherosclerosis. One study (PMID: 34157851) indicates that statins may have inhibitory effects on EndMT in vivo. In this study, 25-OHC increased the expression of TGF-β in HUVECs, while atorvastatin, rosuvastatin and the combination of atorvastatin and ezetimibe decreased this effect. This is contradictory to the anti-inflammatory effect of statins. Whether the effect of atorvastatin and rosuvastatin on TGF-β expression is independent of the anti-inflammatory properties? The authors should discuss it further.

The research we cited, such as Kabel et al. (2017) and Wang et al. (2018) show a correlation of decreased TGFB levels due to statins (lines 233-246). Moreover Chen et al. [42] indicate that inhibition of endothelial TGFβ signaling in hyperlipidemic mice reduces vessel wall inflammation and vascular permeability, resulting in the arrest of disease progression and regression of established lesions. These pro-inflammatory effects resulting from endothelial TGFβ signaling stand in stark contrast with its effects in other cell types, and show the potential of cell-type specific therapeutic intervention aimed at controlling atherosclerosis.

We added the below information in the manuscript: 

Discussion

”Previous studies indicate that TGFβ plays a dual role in atherosclerosis, and that these are related to activation of TGF-β–specific and bone morphogenetic protein (BMP)-specific signaling cascades in macrophages. The inhibition of MMP-2 expression by TGF-beta1 suggests that the latter, acting via SMAD family member 3 (Smad3), may promote plaque stability [40]. Nurgazieva et al. [41] suggest that the effect of TGF-β on atherosclerosis depends on the balance between Smad1/5- and Smad2/3-dependent signaling, where Smad1/5 is pro-atherogenic, and Smad2/3 has anti-atherogenic effects. However, endothelial TGFβ signaling is one of the primary drivers of atherosclerosis-associated vascular inflammation. Chen et al. [42] indicate that inhibition of endothelial TGFβ signaling in hyperlipidemic mice reduces vessel wall inflammation and vascular permeability, resulting in the arrest of disease progression and regression of established lesions. These pro-inflammatory effects resulting from endothelial TGFβ signaling stand in stark contrast with its effects in other cell types, and show the potential of cell-type specific therapeutic intervention aimed at controlling atherosclerosis.” (lines 247-260)

We agree that TGFβ is not suitable to be an anti-inflammatory marker in ECs. Therefore, in addition to the expression of TGFβ, we determined the expression of IL-10. 25-hydroxycholesterol significantly decreased mRNA expression of IL-10 and statins increased it. Therefore, it is not clear whether the effects of atorvastatin and rosuvastatin on TGF-β expression are independent of the anti-inflammatory properties.

3. Previous studies showed that the dual roles of TGFβ in atherosclerosis have been attributed to activation of TGF-β–specific and BMP-specific signaling cascades in macrophages (PMID: 10825169; PMID: 25505291). However, endothelial TGFβ signaling promotes vascular inflammation and atherosclerosis (PMID: 31572976). These contradictory studies indicate that TGFβ is not suitable to be an anti-inflammatory marker in ECs. I suggest that the authors use IL-10 as an anti-inflammatory cytokine in ECs. IL-10 possesses multiple anti-atherogenic activities, including blocking atherogenic lysophosphatidylcholine-induced EC activation (PMID: 31731100) and reducing intercellular adhesion molecule 1 (ICAM-1) expression in ECs (PMID: 12223518).

We agree that TGFβ is not suitable to be an anti-inflammatory marker in ECs. Therefore, in addition to the expression of TGFβ, we determined the expression of IL-10. 

We added the below information in the manuscript: 

Results

25-hydroxycholesterol significantly decreased mRNA expression of IL-10 and IL-37 in HUVECs after 24 hours of culture as compared to unstimulated controls (p<0.05). Atorvastatin (p<0.05) and rosuvastatin (p<0.05; p<0.001), effectively increased mRNA levels of IL-10 and IL37. No significant effect was observed for ezetimibe, ezetimibe-atorvastatin or ezetimibe-rosuvastatin (p>0.05) (Fig. 1D and Fig. 1E). (lines 103-107)

We added the below information in the manuscript: 

Discussion

”It has been preiovusly reported that IL-10 inhibits the proinflammatory activation of endothelial cells (EC) with respect to stimulus and effector response. IL-10 was found to reduce the migration of monocytes and T lymphocytes across the endothelium and decreased endothelial production of chemokines following lipopolysaccharide (LPS) treatment [22]. IL-10 could suppress mitochondrial reactive oxygen species (mtROS), which partially mediate lysophosphatidylcholine-induced (LPC) EC activation and innate immune response [23].” (lines 169-175)

”Previous studies indicate that TGFβ plays a dual role in atherosclerosis, and that these are related to activation of TGF-β–specific and bone morphogenetic protein (BMP)-specific signaling cascades in macrophages. The inhibition of MMP-2 expression by TGF-beta1 suggests that the latter, acting via SMAD family member 3 (Smad3), may promote plaque stability [40]. Nurgazieva et al. [41] suggest that the effect of TGF-β on atherosclerosis depends on the balance between Smad1/5- and Smad2/3-dependent signaling, where Smad1/5 is pro-atherogenic, and Smad2/3 has anti-atherogenic effects. However, endothelial TGFβ signaling is one of the primary drivers of atherosclerosis-associated vascular inflammation. Chen et al. [42] indicate that inhibition of endothelial TGFβ signaling in hyperlipidemic mice reduces vessel wall inflammation and vascular permeability, resulting in the arrest of disease progression and regression of established lesions. These pro-inflammatory effects resulting from endothelial TGFβ signaling stand in stark contrast with its effects in other cell types, and show the potential of cell-type specific therapeutic intervention aimed at controlling atherosclerosis.” (lines 247-260)

4. In the method, HUVECs were treated with oxysterol for 4 hours, then washed out and the cells were stimulated with the corresponding drugs. Is there any reason to wash oxysterol out before addition of a drug under study? The authors shall co-incubate cells with oxysterol and other drugs throughout the duration of co-treatment.

We thank the Reviewer for this valuable suggestion. 

We added in the manuscript this information: 

Discussion

We added the below information in the manuscript: 

”High-fat meals (are rich with the triglycerides (TG) and oxysterols: 25-hydroxycholesterol, 7 a-hydroxycholesterol, 7 b-hydroxycholesterol, a-epoxycholesterol, b-epoxycholesterol and 7-ketocholesterol) can impair endothelial function within three to four hours, which is associated with peak postprandial lipemia [17]. In healthy people, plasma TG levels peak three to four hours after the ingestion of a high-fat meal, and tend to return to baseline within six to eight hours [18]. Therefore, 25-OHC stimulation was stopped after four hours. ’’ (lines 157-163)

Moreover, we used the same research model in our previous works: doi: 10.1016/j.biopha.2022.112679; doi: 10.3390/ijms21061953.10.1080/21688370.2021.1956284; doi: 10.1371/journal.pone.0256996.

Minor

1. It should be highlighted that this manuscript requires rigorous editing by someone with competence in technical English grammar and sentence structure so that the aims and results are clear to the reader.

English of the manuscript was carefully checked and corrected by an English native speaker (see the attached certificate).

 

Reviewer #2: PLOS One Review 07/12/22 - PONE-D-22-07586

1. This article presents cytokine gene expression data and IL-35 concentrations from HUVECs stimulated with 25-OH-cholesterol that were treated or not with statins and/or ezetimibe. The authors conclude that there is no advantage of adding ezetimibe to statins over statin monotherapy. The manuscript addresses an important topic of interest. However, based on the in vitro experiments presented in this manuscript, this conclusion does not seem to be justified. The fact that ezetimibe in combination with statins under the specific experimental conditions does not further decrease the gene expression of pro-inflammatory cytokines in HUVECs, does not proof that such combinations are without any benefits to other tissues and cells relevant for arteriosclerosis formation. Ezetimibe inhibits intestinal cholesterol absorption, which in vivo prevents the inflammatory and damaging effects of cholesterol on vascular endothelium. While this study demonstrates that the anti-inflammatory effects of ezetimibe are minimal and do not enhance the anti-inflammatory effects of statins when added in combination, no statements regarding clinical benefits can be drawn from this study. The conclusions therefore should be changed accordingly, and the limitations of this in vitro study should be emphasized in the Discussion section of this manuscript. Furthermore, the study from Oh et al. (reference 35) compares high-dose rosuvastatin vs low-dose rosuvastatin combined with ezetimibe. Therefore, no group differences might indicate an advantage of the combination therapy, since lower doses may reduce adverse treatment effects. The statement in lines 248 to 250 seems therefore incorrect.

We thank the Reviewer for this valuable suggestion. We changed the conclusions and the limitations of this study were emphasized in the Discussion section of this manuscript. The statement in lines 248 to 250 has been removed.

We added the below information in the manuscript: 

Discussion

The main limitation of our work is that it is not possible to directly translate our in vitro findings into the process of atherosclerosis taking place in vivo. Our research suggests that, under these specific experimental conditions, ezetimibe in combination with statins does not further decrease the gene expression of pro-inflammatory cytokines in endothelial cells stimulated by 25-OHC in vitro; however, this does not mean that such combinations are without any benefits to other tissues and cells relevant for arteriosclerosis formation in vivo. Ezetimibe inhibits the absorption of cholesterol from the small intestine, leading to a reduction in intestinal cholesterol transmission to the liver, which protects the on vascular endothelium from the inflammatory and damaging effects of cholesterol in vivo [43]. Although our study showed that the anti-inflammatory effects of ezetimibe were minimal and did not enhance the anti-inflammatory effects of statins when added in combination, it is not possible to draw firm statements regarding clinical benefits. Therefore, further studies are needed to compare statin monotherapy with statin-ezetimibe combined therapy in vivo. (lines 302-314).

2. The Method description is unclear and inconsistent with the statements in the Abstract and Results section of the manuscript. For example, the duration of culture of HUVECs varies throughout the manuscript between 12 and 24 hours. 

We thank the Reviewer. We corrected the incorrect description.

Materials and methods

Cells treatment in the viability level and gene expression

Gene expression

Enzyme-linked immunosorbent assay (ELISA) for IL-35

mRNA and supernatants were isolated after 20 hours of drug incubation, which corresponds to 24 hours of the experiment (four hours of pre-incubation with 25-OHC + 20 hours of drug incubation).

Figure 1 A-F. Relative expression of genes (A-F) in HUVECs prestimulated with 25-OHC (10 μg/mL) for four hours, followed by removal of oxysterol and stimulation for another 20 hours (i.e. 24-hours in total) with rosuvastatin (10 μM; 4815 ng/mL), atorvastatin (5 μM; 2793 ng/mL), ezetimibe (500 ng/mL), ezetimibe-rosuvastatin and ezetimibe-atorvastatin. Each experimental group has a control group that was not pre-stimulated with 25-OHC. Mean ± SD calculated from four biological replicates. Significant differences from negative controls are indicated by *P < 0.05; **P < 0.01; ***P < 0.001. Statistical analysis was conducted using one-way ANOVA and post hoc Tukey’s test.

Figure 2 A-C. Relative expression of gene IL-35 (EBI3-A, IL12-B) and protein (C) in HUVECs prestimulated with 25-OHC (10 μg/mL) for four hours of incubation, followed by removal of oxysterol and stimulation for another 20 hours (i.e. 24-hour incubation in total) with rosuvastatin (10 μM; 4815 ng/mL), atorvastatin (5 μM; 2793 ng/mL), ezetimibe (500 ng/mL), ezetimibe plus rosuvastatin and ezetimibe plus atorvastatin. Each experimental group has a control group without pre-stimulation of 25-OHC. Mean ± SD calculated from four biological replicates. Significant differences from negative controls are indicated by *P < 0.05; **P < 0.01; ***P < 0.001. Statistical analysis was conducted using one-way ANOVA and post hoc Tukey’s test.

3. Furthermore, the number of repetitions of the experiments remains unclear. 

We thank the Reviewer for this valuable suggestion. This sentence has been corrected.

We added the below information in the manuscript: 

Statistical analysis

”Each analysis was performed in four independent experiments, with each experiment being repeated three times.” (lines 383-385).

4. And the current manuscript version contains a number of grammar and spelling errors. 

English of the manuscript was carefully checked and corrected by an English native speaker (see the attached certificate).

5. Abbreviations are not consistently defined upon their first use in the manuscript.

We thank the Reviewer. Abbreviations were defined upon their first use in the manuscript.

6. IL-35 is called anti-inflammatory in the abstract, but pro-inflammatory in the introduction and other parts of the manuscript.

We thank the Reviewer for this valuable suggestion. This introductory paragraph was rephrased.

“A detailed analysis of the expression of pro-inflammatory IL-1�, IL-18, IL-23, and anti-inflammatory IL35 (subunits IL12A, EBI3), transforming growth factor β (TGF-�), IL-10 and IL-37 was performed at the mRNA level. Additionally, IL-35 expression was determined at the protein level. “ (lines 62-66).

We added in the manuscript this information: 

Discussion

”Previous studies indicate that IL-35 is a responsive anti-inflammatory cytokine, which is upregulated during inflammatory response, but not a constitutively expressed housekeeping cytokine. IL-35 is not constitutively expressed in tissues, being induced by pro-inflammatory stimuli, and inhibits lipopolysaccharide (LPS)-induced endothelial cell activation [27]. ” (lines 179-183).

”Furthermore, previous data suggest that IL-35 can suppress endothelial cell activation by inhibiting mtROS generation. Indeed, mtROS is known to contribute to inflammatory cytokine production and innate immune responses in macrophages and T cells [27]. (lines 211-214).

”Atorvastatin was found to inhibit dihydronicotinamide-adenine dinucleotide phosphate (NADPH)-oxidase dependent reactive oxygen species (ROS) generation; however, it showed no effect on mtROS generation, and activated IL-1β and IL-6 secretion in peripheral blood mononuclear cells (PBMC) from control and type 2 diabetes (T2D) patients. In turn, rosuvastatin mitigates coronary microembolization (CME)-induced cardiac injury by inhibiting NADPH-oxidase (Nox2)-induced ROS overproduction and alleviating apoptosis-related protein (p53/Bax/Bcl-2)-dependent cardiomyocyte apoptosis [35]. ’’ (lines 2515-221).

Complete the literature reference:

[27] Li X, Shao Y, Sha X, Fang P, Kuo YM, Andrews AJ, et al. IL-35 (interleukin-35) suppresses endothelial cell activation by inhibiting mitochondrial reactive oxygen species-mediated site-specific acetylation of H3K14 (histone 3 lysine 14). Arterioscler Thromb Vasc Biol. 2018;38: 599–609. doi:10.1161/ATVBAHA.117.310626

[35] Cao Y, Chen Z, Jia J, Chen A, Gao Y, Qian J, Ge J. Rosuvastatin Alleviates Coronary Microembolization-Induced Cardiac Injury by Suppressing Nox2-Induced ROS Overproduction and Myocardial Apoptosis. Cardiovasc Toxicol. 2022 Apr;22(4):341-351. doi: 10.1007/s12012-021-09716-4.

7. Lines 77-79: “mRNA and supernatants were isolated after the 20 hours of drug stimulation. mRNA and supernatants were then isolated from HUVECs after 25-hydroxycholesterol (25-OHC) pre-stimulation and drug stimulation.” This seems to be duplicate statements.

We thank the Reviewer, one sentence has been deleted.

mRNA and supernatants were isolated after the 20 hours of drug stimulation. 

We added the below information in the manuscript: 

Following 25-OHC pre-stimulation and drug stimulation, mRNA and the supernatants were isolated from HUVECs

Line 94, similar lines 97-98: “…was added after 25-OHC prestimulation reduced the IL-18 mRNA level”. This seems to be a grammar error.

We thank the Reviewer. This sentence has been corrected. Atorvastatin (p<0.001), rosuvastatin (p<0.05), and atorvastatin-ezetimibe (p<0.01) or rosuvastatin-ezetimibe (p<0.001) added after 25-OHC prestimulation also reduced IL-18 mRNA levels.

8. Line 138: “protine”. Please change to “protein”.

We changed.

9. Please add a reference for the statement in lines 154-156

They also correspond to those in the plasma of patients receiving the highest therapeutic doses of these drugs: 80 mg for atorvastatin [13], 40 mg for rosuvastatin [14], and 10 mg for ezetimibe [15,16]. (lines 154-156).

13. Giordano A, Romano S, Monaco M, Sorrentino A, Corcione N, di Pace AL, et al. Differential effect of atorvastatin and tacrolimus on proliferation of vascular smooth muscle and endothelial cells. Am J Physiol - Hear Circ Physiol. 2012;302: 135–143. doi:10.1152/ajpheart.00490.2011

14. Padró T, Lugano R, García-Arguinzonis M, Badimon L. LDL-induced impairment of human vascular smooth muscle cells repair function is reversed by HMG-CoA reductase inhibition. PLoS One. 2012;7. doi:10.1371/journal.pone.0038935

15. Becher T, Schulze TJ, Schmitt M, Trinkmann F, El-Battrawy I, Akin I, et al. Ezetimibe inhibits platelet activation and uPAR expression on endothelial cells. Int J Cardiol. 2017;227: 858–862. doi:10.1016/j.ijcard.2016.09.122

16. Kosoglou T, Statkevich P, Johnson-Levonas AO, Paolini JF, Bergman AJ, Alton KB. Ezetimibe: A review of its metabolism, pharmacokinetics and drug interactions. Clin Pharmacokinet. 2005;44: 467–494. doi:10.2165/00003088-200544050-00002

10. Lines 162-164: “Further, the role of IL-37 may be related to the IL-18 pathway extracellularly and involved in the adhesion and transmigration of neutrophils in human coronary artery endothelial cells (HCAECs).” Please correct the sentence structure.

We thank the Reviewer. This sentence has been corrected.

Furthermore, extracellular IL-37 may influence the IL-18 pathway, as well as the adhesion and transmigration of neutrophils in human coronary artery endothelial cells (HCAECs) [25].

11.Lines 220: “for other analyzing genes”. Did you mean “for the other analyzed genes”?

Yes, of course. We thank the Reviewer. This error has been corrected.

12. Lines 239-240: ‘aniinflammatory effect”. Please correct to “anti-inflammatory effect”

We changed.

13. Line 241: What does the abbreviation “hsCRP” stand for?

We thank the Reviewer. Abbreviations were defined upon their first use in the manuscript.

high-sensitivity C-reactive protein (hsCRP)

14. Line 252: “Our study shows that 25-hydroxycholesterol can damage the endothelium …”. Cytokine gene expression changes alone do not proof any endothelial damage.

We thank the Reviewer for this valuable suggestion. This sentence has been corrected. “It appears that 25-hydroxycholesterol can induce inflammation in endothelial cells.” (Pages , lines ).

15. Line 263: “purchased in Lonza”. Please change to “purchased from Lonza”. Similar errors were repeated several times in the manuscript text.

We thank the Reviewer for this valuable suggestion. We changed this.

16 Line 265: “Rneasy Mini Kit”. Please change to “RNeasy Mini Kit”. We changed.

17. Line 270: “Cells treatment in the viability level and gene expression” This rephrase the title.

We thank the Reviewer for this valuable suggestion. We changed the title to: “Cell culture in monolayers”

18. Line 279: What is “proper medium”?

We thank the Reviewer for this valuable suggestion. This sentence has been corrected.

We added the below information in the manuscript: 

”Both trypsinized HUVECs were separately seeded on 24-well plates at a density of 100,000 cells per well in a 600 µl EGM-2.” 

19. Could the authors pleased explain why 25-hydroxycholesterol was removed prior to treatment with statins and/or ezetimibe, since in vivo high cholesterol levels would be present throughout the treatment with these drugs.

We added the below information in the manuscript: 

”High-fat meals (are rich with the triglycerides (TG) and oxysterols: 25-hydroxycholesterol, 7 a-hydroxycholesterol, 7 b-hydroxycholesterol, a-epoxycholesterol, b-epoxycholesterol and 7-ketocholesterol) can impair endothelial function within three to four hours, which is associated with peak postprandial lipemia [17]. In healthy people, plasma TG levels peak three to four hours after the ingestion of a high-fat meal, and tend to return to baseline within six to eight hours [18]. Therefore, 25-OHC stimulation was stopped after four hours. ’’ (lines 157-163)

Moreover, we used the same research model in our previous works: doi: 10.1016/j.biopha.2022.112679; doi: 10.3390/ijms21061953.10.1080/21688370.2021.1956284; doi: 10.1371/journal.pone.0256996.

Line 292: “G Gene expression”. Did the authors mean “Gene expression”?

Yes, of course. We thank the Reviewer. This error has been corrected.

Line 301: Livak and Schmittgen 2001: This reference is not included in the reference list.

We thank the Reviewer. This error has been corrected.

[49] Livak KJ, Schmittgen TD. Analysis of relative gene expression data using real-time quantitative PCR and the 2-ΔΔCT method. Methods. 2001;25: 402–408. doi:10.1006/meth.2001.1262

Line 317: “Shapiro-Will test”. Please correct to “Shapiro-Wilk test”.

We thank the Reviewer. This error has been corrected.

Lines 320-322: “The individual analysis was performed in nine-four independent experiments, while each experiment was repeated twice or three times depending on the method.” This description seems unclear.

We thank the Reviewer for this valuable suggestion. This sentence has been corrected.

We added the below information in the manuscript: 

Statistical analysis

” Each analysis was performed in four independent experiments, with each experiment being repeated three times.”

---

## [Decision Letter · Decision Letter 1]

8 Nov 2022

PONE-D-22-07586R1Effect of lipid-lowering therapies on pro-inflammatory and anti-inflammatory properties of vascular endothelial cellsPLOS ONE

Dear Dr Wozniak 

Thank you for submitting your manuscript to PLOS ONE. After careful consideration, we feel that it has merit but does not fully meet PLOS ONE’s publication criteria as it currently stands. Therefore, we invite you to submit a revised version of the manuscript that addresses the points raised during the review process.

We look forward to receiving your revised manuscript.

Kind regards,

Xian Wu Cheng, M.D., Ph.D., FAHA

Academic Editor

PLOS ONE

Journal Requirements:

Reviewers' comments:

Reviewer's Responses to Questions

**Comments to the Author**

1. If the authors have adequately addressed your comments raised in a previous round of review and you feel that this manuscript is now acceptable for publication, you may indicate that here to bypass the “Comments to the Author” section, enter your conflict of interest statement in the “Confidential to Editor” section, and submit your "Accept" recommendation.

Reviewer #1: All comments have been addressed

Reviewer #2: (No Response)

2. Is the manuscript technically sound, and do the data support the conclusions?

Reviewer #1: Yes

Reviewer #2: Yes

3. Has the statistical analysis been performed appropriately and rigorously? 

Reviewer #1: Yes

Reviewer #2: Yes

4. Have the authors made all data underlying the findings in their manuscript fully available?

Reviewer #1: Yes

Reviewer #2: No

5. Is the manuscript presented in an intelligible fashion and written in standard English?

Reviewer #1: Yes

Reviewer #2: Yes

6. Review Comments to the Author

Reviewer #1: The authors have addressed most of the questions. However, this manuscript still needs careful editing regarding English grammar and sentence structure. For example, in line 174, please change “lysophosphatidylcholine-induced (LPC)” to “lysophosphatidylcholine (LPC)-induced”. In line 182, please delete the sentence “IL-35 is not constitutively expressed in tissues,”, which is duplicated with the last sentence. In line 198, please change “bene” to “been”.

Reviewer #2: The revised manuscript is significantly improved and all reviewer questions have been addressed. There are a couple of remaining regarding the last sentence of the abstract and one statement in the method section of the revised manuscript.

Abstract: “The anti-inflammatory effect of the combination therapies appears to be based on the effects of the statins alone and not their combination with ezetimibe; therefore, they offer no advantage over statin monotherapy.” The last part of this statement seems incorrect, since the fact that combination therapies with statins and ezetimibe do not provide enhanced anti-inflammation in HUVECs over statin monotherapy does not mean that they offer no advantage.

Page 14: “After reaching 80–90 % conﬂuence, HUVECs were stimulated with 25-hydroxycholesterol (10 µg/ml) for four hours. After incubation, the cells were centrifuged, the compound was discarded, and HUVECs were stimulated with atorvastatin (5 µM), rosuvastatin (10 µM) and ezetimibe (500 ng/ml) for 24 hours. After incubation, the cells were centrifuged, the compounds were discarded, and the cells were resuspended in EGM-2 medium.”

This description appears confusing and the timing should probably read 20 hours instead of 24 hours.

Since PLoS ONE does not provide proof reading, please revise the following spelling and grammar errors as outlines below:

Page 3: “Statins, a class of cholesterol-lowering medications - low-density lipoprotein (LDL) that inhibit 3-hydroxy-3-methyl-glutaryl-coenzyme A reductase. This drugs are commonly administered to treat atherosclerotic cardiovascular disease (CVD).” Please correct these sentences, which seem grammatically incorrect.

Page 4: “…Analysis of the expression of genes pro-inflammatory cytokines”. Please change to “Analysis of the expression of genes of pro-inflammatory cytokines”. Likewise for page 5:“…Analysis of the expression of genes anti--inflammatory cytokines”

Page 4: “The administration of 25-hydroxycholesterol was associated with an increase in IL-18 mRNA (p<0.01).” According to the corresponding figure, p should be < 0.001.

Page 7: “The present study focuses on key inflammatory factors that play important role in the endothelium [19].” Please change to “…that play an important role…”

Page 7: Please correct the spelling for “preiovusly”.

Page 8: “Previous studies indicate that IL-35 is a responsive anti-inflammatory cytokine, which is upregulated during inflammatory response, but not a constitutively expressed housekeeping cytokine. IL-35 is not constitutively expressed in tissues, being induced by pro-inflammatory stimuli, and inhibits lipopolysaccharide (LPS)-induced endothelial cell activation [27].”

Please change to “Previous studies indicate that IL-35 is a responsive anti-inflammatory cytokine, which is upregulated during inflammatory responses, but not a constitutively expressed housekeeping cytokine. IL-35 is not constitutively expressed in tissues, but is induced by pro-inflammatory stimuli, and inhibits lipopolysaccharide (LPS)-induced endothelial cell activation [27].”

Page 8: “In turn TGF-β, plays a key role in proper cell activity and is widely considered an anti-inflammatory cytokine.”

Please change to: “In turn TGF-β, which plays a key role in proper cell activity, is widely considered an anti-inflammatory cytokine.”

Page 8: “Atorvastatin and rosuvastatin, have…”. Please remove the comma.

Page 8: Please be consistent in the tenses used. For example “Zhang et al. (2018) [29] report that the administration of atorvastatin (5 mg/kg/day) was associated with a reduction of IL-1β in the serum of rabbits. Furthermore, Satoh et al. (2014) [30] presented…”

Page 8: “…in human monocytic cells line derived from an acute monocytic leukemia patients…” Please change to “…in human monocytic cell lines derived from an acute monocytic leukemia patient…”

Page 8 to 9: Please correct the spelling of “It has bene reported …”

Page 9: “…can reduce IL-23 serum level” Please change to “…can reduce IL-23 serum levels”.

Page 9: “…is in line with the Shin et al. (2017) [34].” Please change to “…is in line with Shin et al. (2017) [34].”

Page 10: Please correct the spelling of “tool-like receptor 2”.

Page 10: Please change “…IL-6 a in trastuzumab-treated mice” to “IL-6 in trastuzumab-treated mice”.

Page 12: Please correct the spelling of “Apopoliprotein E (ApoE)”.

Page 13: Please change “…which protects the on vascular endothelium…” to “…which protects the vascular endothelium…”.

Page 14: Is the following correct? “…human recombinant E.coli, lyophilized powder (R3-IGF-1)…”

Page 14: Please change “…gentamicin (GA-1000), and heparin, fetal bovine serum (FBS), …” to “… gentamicin (GA-1000), heparin, and fetal bovine serum (FBS), …”.

Page 14: Please change “Atorvastatin, rosuvastatin, ezetimibe, 25-hydroxycholesterol (25-OHC) and using primers were bought from Sigma-Aldrich (USA).” To “Atorvastatin, rosuvastatin, ezetimibe, 25-hydroxycholesterol (25-OHC) and primers were bought from Sigma-Aldrich (USA).”

Please 14: Please change “RNeasy Mini Kit was bought in Qiagen (Germany).” To “RNeasy Mini Kit was bought from Qiagen (Germany).”

Page 14: Please correct this sentence “Both trypsinized HUVECs were separately seeded on 24-well plates at a density of 100,000 cells per well in a 600 µlEGM-2.”

Page 18 reference 13: Please correct the abbreviated name of the cited journal.

7. PLOS authors have the option to publish the peer review history of their article (what does this mean?). If published, this will include your full peer review and any attached files.

Reviewer #1: No

Reviewer #2: No

---

## [Author Response · Author response to Decision Letter 1]

19 Dec 2022

The suggestions and questions raised by Reviewer #1

Reviewer #1: The authors have addressed most of the questions. However, this manuscript still needs careful editing regarding English grammar and sentence structure. For example, in line 174, please change “lysophosphatidylcholine-induced (LPC)” to “lysophosphatidylcholine (LPC)-induced”. In line 182, please delete the sentence “IL-35 is not constitutively expressed in tissues,”, which is duplicated with the last sentence. In line 198, please change “bene” to “been”.

We thank the Reviewer for this valuable suggestion. English of the manuscript was carefully checked and corrected once more by an English native speaker (please see the attached certificate). 

The suggestions and questions raised by Reviewer #2

Reviewer #2: The revised manuscript is significantly improved and all reviewer questions have been addressed. There are a couple of remaining regarding the last sentence of the abstract and one statement in the method section of the revised manuscript.

Abstract: “The anti-inflammatory effect of the combination therapies appears to be based on the effects of the statins alone and not their combination with ezetimibe; therefore, they offer no advantage over statin monotherapy.” The last part of this statement seems incorrect, since the fact that combination therapies with statins and ezetimibe do not provide enhanced anti-inflammation in HUVECs over statin monotherapy does not mean that they offer no advantage.

We thank the Reviewer for this valuable suggestion. The sentence has been removed from the abstract.

Page 14: “After reaching 80–90 % conﬂuence, HUVECs were stimulated with 25-hydroxycholesterol (10 µg/ml) for four hours. After incubation, the cells were centrifuged, the compound was discarded, and HUVECs were stimulated with atorvastatin (5 µM), rosuvastatin (10 µM) and ezetimibe (500 ng/ml) for 24 hours. After incubation, the cells were centrifuged, the compounds were discarded, and the cells were resuspended in EGM-2 medium.” This description appears confusing and the timing should probably read 20 hours instead of 24 hours.

We thank the Reviewer for this valuable suggestion. We changed the description:

After reaching 80–90 % conﬂuence, HUVECs were stimulated with 25-hydroxycholesterol (10 µg/ml) for four hours. After incubation, the cells were centrifuged, the compound was discarded, and HUVECs were stimulated with atorvastatin (5 µM), rosuvastatin (10 µM) and ezetimibe (500 ng/ml) for 20 hours. Total cell stimulation was 24 hours: four hours of pre-incubation with 25-OHC + 20 hours of drug incubation. After incubation, the cells were centrifuged, the compounds were discarded, and the cells were resuspended in EGM-2 medium.”

Since PLoS ONE does not provide proof reading, please revise the following spelling and grammar errors as outlines below:

Page 3: “Statins, a class of cholesterol-lowering medications - low-density lipoprotein (LDL) that inhibit 3-hydroxy-3-methyl-glutaryl-coenzyme A reductase. This drugs are commonly administered to treat atherosclerotic cardiovascular disease (CVD).” Please correct these sentences, which seem grammatically incorrect.

We corrected this sentence:

The statins are cholesterol-lowering medications that inhibit the activity of 3-hydroxy-3-methyl-glutaryl-coenzyme A reductase. These drugs are commonly administered to treat atherosclerotic cardiovascular disease (CVD).

Page 4: “…Analysis of the expression of genes pro-inflammatory cytokines”. Please change to “Analysis of the expression of genes of pro-inflammatory cytokines”. Likewise for page 5:“…Analysis of the expression of genes anti--inflammatory cytokines”.

We have corrected these sentences. Analysis of the expression of genes of pro-inflammatory cytokines and Analysis of the expression of genes anti--inflammatory cytokines

Page 4: “The administration of 25-hydroxycholesterol was associated with an increase in IL-18 mRNA (p<0.01).” According to the corresponding figure, p should be < 0.001.

We thank the Reviewer for this valuable suggestion, we have corrected the error.

Page 7: “The present study focuses on key inflammatory factors that play important role in the endothelium [19].” Please change to “…that play an important role…”

We thank the Reviewer for this valuable suggestion, we have corrected the error.

Page 7: Please correct the spelling for “previously”.

We thank the Reviewer for this valuable suggestion, we have corrected the error.

Page 8: “Previous studies indicate that IL-35 is a responsive anti-inflammatory cytokine, which is upregulated during inflammatory response, but not a constitutively expressed housekeeping cytokine. IL-35 is not constitutively expressed in tissues, being induced by pro-inflammatory stimuli, and inhibits lipopolysaccharide (LPS)-induced endothelial cell activation [27].”

Please change to “Previous studies indicate that IL-35 is a responsive anti-inflammatory cytokine, which is upregulated during inflammatory responses, but not a constitutively expressed housekeeping cytokine. IL-35 is not constitutively expressed in tissues, but is induced by pro-inflammatory stimuli, and inhibits lipopolysaccharide (LPS)-induced endothelial cell activation [27].”

We thank the Reviewer for this valuable suggestion, we have corrected the error.

Page 8: “In turn TGF-β, plays a key role in proper cell activity and is widely considered an anti-inflammatory cytokine.”

Please change to: “In turn TGF-β, which plays a key role in proper cell activity, is widely considered an anti-inflammatory cytokine.”

We thank the Reviewer for this valuable suggestion, we have corrected the error.

Page 8: “Atorvastatin and rosuvastatin, have…”. Please remove the comma.

We thank the Reviewer for this valuable suggestion, we have corrected the error.

Page 8: Please be consistent in the tenses used. For example “Zhang et al. (2018) [29] report that the administration of atorvastatin (5 mg/kg/day) was associated with a reduction of IL-1β in the serum of rabbits. Furthermore, Satoh et al. (2014) [30] presented…”

We thank the Reviewer for this valuable suggestion, we have corrected the error.

Page 8: “…in human monocytic cells line derived from an acute monocytic leukemia patients…” Please change to “…in human monocytic cell lines derived from an acute monocytic leukemia patient…”

We thank the Reviewer for this valuable suggestion, we have corrected the error.

Page 8 to 9: Please correct the spelling of “It has bene reported …”

We thank the Reviewer for this valuable suggestion, we have corrected the error.

Page 9: “…can reduce IL-23 serum level” Please change to “…can reduce IL-23 serum levels”.

We thank the Reviewer for this valuable suggestion, we have corrected the error.

Page 9: “…is in line with the Shin et al. (2017) [34].” Please change to “…is in line with Shin et al. (2017) [34].”

We thank the Reviewer for this valuable suggestion, we have corrected the error.

Page 10: Please correct the spelling of “tool-like receptor 2”.

We thank the Reviewer for this valuable suggestion, we have corrected the error.

Page 10: Please change “…IL-6 a in trastuzumab-treated mice” to “IL-6 in trastuzumab-treated mice”.

We thank the Reviewer for this valuable suggestion, we have corrected the error.

Page 12: Please correct the spelling of “Apopoliprotein E (ApoE)”.

We thank the Reviewer for this valuable suggestion, we have corrected the error.

Page 13: Please change “…which protects the on vascular endothelium…” to “…which protects the vascular endothelium…”.

We thank the Reviewer for this valuable suggestion, we have corrected the error.

Page 14: Is the following correct? “…human recombinant E.coli, lyophilized powder (R3-IGF-1)…”

The following statement is correct.

Page 14: Please change “…gentamicin (GA-1000), and heparin, fetal bovine serum (FBS), …” to “… gentamicin (GA-1000), heparin, and fetal bovine serum (FBS), …”.

We have corrected the error.

Page 14: Please change “Atorvastatin, rosuvastatin, ezetimibe, 25-hydroxycholesterol (25-OHC) and using primers were bought from Sigma-Aldrich (USA).” To “Atorvastatin, rosuvastatin, ezetimibe, 25-hydroxycholesterol (25-OHC) and primers were bought from Sigma-Aldrich (USA).”

We have corrected the error.

Please 14: Please change “RNeasy Mini Kit was bought in Qiagen (Germany).” To “RNeasy Mini Kit was bought from Qiagen (Germany).”

We have corrected the error.

Page 14: Please correct this sentence “Both trypsinized HUVECs were separately seeded on 24-well plates at a density of 100,000 cells per well in a 600 µlEGM-2.”

We have corrected the error.

Page 18 reference 13: Please correct the abbreviated name of the cited journal.

We have corrected the error.

---

## [Decision Letter · Decision Letter 2]

8 Jan 2023

The effect of lipid-lowering therapies on the pro-inflammatory and anti-inflammatory properties of vascular endothelial cells

PONE-D-22-07586R2

Dear Dr Wozniak

We’re pleased to inform you that your manuscript has been judged scientifically suitable for publication and will be formally accepted for publication once it meets all outstanding technical requirements.

Kind regards,

Xian Wu Cheng, M.D., Ph.D., FAHA

Academic Editor

PLOS ONE

Additional Editor Comments (optional):

Although the original reviewer#2 has declined to review the second round (as minor revision in the first rund), all original concerns have addressed by the authors.

Reviewers' comments:

Reviewer's Responses to Questions

**Comments to the Author**

1. If the authors have adequately addressed your comments raised in a previous round of review and you feel that this manuscript is now acceptable for publication, you may indicate that here to bypass the “Comments to the Author” section, enter your conflict of interest statement in the “Confidential to Editor” section, and submit your "Accept" recommendation.

Reviewer #1: All comments have been addressed

2. Is the manuscript technically sound, and do the data support the conclusions?

Reviewer #1: Yes

3. Has the statistical analysis been performed appropriately and rigorously? 

Reviewer #1: Yes

4. Have the authors made all data underlying the findings in their manuscript fully available?

Reviewer #1: Yes

5. Is the manuscript presented in an intelligible fashion and written in standard English?

Reviewer #1: Yes

6. Review Comments to the Author

Reviewer #1: (No Response)

7. PLOS authors have the option to publish the peer review history of their article (what does this mean?). If published, this will include your full peer review and any attached files.

Reviewer #1: No

---

## [Editor Report · Acceptance letter]

16 Jan 2023

PONE-D-22-07586R2 

The effect of lipid-lowering therapies on the pro-inflammatory and anti-inflammatory properties of vascular endothelial cells 

Dear Dr. Woźniak:

I'm pleased to inform you that your manuscript has been deemed suitable for publication in PLOS ONE. Congratulations! Your manuscript is now with our production department. 

Kind regards, 

on behalf of

Professor Xian Wu Cheng 

Academic Editor

PLOS ONE